# Tuberculosis Mortality in Children under Fifteen Years of Age: Epidemiological Situation in Colombia, 2010–2018

**DOI:** 10.3390/tropicalmed7070117

**Published:** 2022-06-25

**Authors:** Laura Plata-Casas, Leandro González-Támara, Favio Cala-Vitery

**Affiliations:** Doctorate in Policy Modeling and Public Management, Faculty of Natural Sciences and Engineering, University Jorge Tadeo Lozano, Bogota 111711, Colombia; leandro.gonzalez@utadeo.edu.co (L.G.-T.); favio.cala@utadeo.edu.co (F.C.-V.)

**Keywords:** potential years of life lost, tuberculosis, Colombia

## Abstract

Pediatric tuberculosis is a serious infectious disease and a hidden global epidemic. The objective of this study was to describe the epidemiological situation of tuberculosis mortality in children under 15 years of age in Colombia in the period 2010–2018. A longitudinal descriptive study was conducted. The variables sex, age groups, and origin were studied. This study had 260 cases for analysis and was carried out in three phases. The first phase was the determination of the sociodemographic and clinical characteristics. The second phase was the construction of indicators by territorial entities. The third phase was stratification into four epidemiological situations according to the mortality rate and years of life lost. The median age was 7 years (range 0–14), 66.5% of cases were pulmonary tuberculosis (97.7% without bacteriological confirmation), 14.3781 years of life lost were recorded (95% CI: 142.811–168.333), and in children under 10–14 years, the loss was 110,057. Amazonas had the highest adjusted YLL rate (3979.7). In total, 36.4% of the territories had a high mortality, and 30.3% adjusted to the situation designated as 1. This is the first study that has used composite indicators to address the problem of premature mortality from childhood tuberculosis in Colombia. Our results allow us to specify that this disease remains a challenge for public health. It requires models of care and differential strategies by region. It also requires ensuring opportunities in diagnosis with sensitive methods, as well as intersectoral work for the optimal approach.

## 1. Introduction

Tuberculosis (TB) is the most important infectious disease in the world and one of the top 10 causes of death [1]. According to data provided by the World Health Organization, tuberculosis is present worldwide, it is the thirteenth cause of death, and the deadliest infectious disease behind COVID-19 (above HIV/AIDS). Approximately 2.2 million deaths were reported among school-age children and young people in 2019. In addition, it is estimated that one-third of the world’s population is infected by the bacillus of Koch. In the last four decades, with the appearance of the human immunodeficiency virus (HIV) infection cases have increased, and new multidrug-resistant strains have appeared which has caused the re-emergence of this event in the health care setting as well as the potentiation of its lethality. Children should be considered a priority in the global approach to this disease [2,3]. The risk of developing tuberculosis among exposed infants and young children is high [4], and due to gaps in diagnosis and access to care, it is associated with a high mortality rate [5].

Investigating children suspected of having TB is difficult given the paucibacillary nature of the disease, the varied clinical manifestations, the gaps in diagnosis [6], and the geographical and administrative barriers that delay diagnosis and treatment [7]. Although there are several challenges and controversies in the approach to pediatric tuberculosis [8], the need to advocate for a screening approach [9,10] has been demonstrated to identify, examine, and monitor children who have been exposed to tuberculosis [11], initiate their treatment, and prevent their death.

Global statistics estimate that in 2020, 1.4 million deaths from tuberculosis were recorded worldwide. Children bear a substantial burden of tuberculosis; annually 12% of those <15 years old become sick, and 16% die. One in four children suffering from tuberculosis will die from this disease [8]; most of these children do not receive timely detection and treatment, and about 21% of children under five years of age received “preventive therapy” in 2019 [3].

In Latin American countries, the behavior of TB has presented a slow increase in incidence and a slow decrease in mortality (1.8% per year), contributing 2.9% of cases worldwide and 290,000 deaths [3], respectively. Colombia has a fluctuating incidence with an increasing tendency; the 2018 mortality rate was 2.0 cases/100,000 inhabitants, with subnational differences between 0.5 and 4.5/100,000 inhabitants, which led to Colombia being classified as a high-burden country [12] and is among the countries whose mortality rate (er 100,000 inhabitants) ranges between 0 and 10 [8].

The years of life lost (YLL), an indicator of premature mortality that allows the study of health inequalities, is estimated by the Global Burden of Disease study to support public health planning and govern health policies and programs at the global, national, and subnational levels [13,14], and guide public health actions [13]. Epidemiological risk stratification (ERE) is part of the epidemiological risk approach in the integrated diagnosis–intervention–evaluation process which methodologically categorizes and integrates geoecological areas according to risk factors, obtaining an objective diagnosis to carry out preventive planning and control of the different diseases [15].

Colombia is located in South America; it is divided into 32 departments and has about 51 million inhabitants as of 2022, 51.2% of whom are women. In its territory, there are important internal and external migratory phenomena, while its legal economy is diversified and is based mainly on agriculture, livestock, oil, trade, and industry.

In Colombia, there have been few studies carried out to determine the mortality from tuberculosis in the child population. Based on the application of the YLL methodology, using the national population projections for the years 2005–2020 of the National Department of Statistics and according to the methodology of the epidemiological risk stratification of the strategic plan “Colombia free of tuberculosis, 2010–2015”, the objective was to describe the epidemiological situation of mortality from tuberculosis in Colombia in children under 15 years of age as well as stratifying its departments by epidemiological situations according to the rates of YLL during the period 2010–2018 and establishing the relationship between these rates, sociodemographic variables, sex, and age group.

The results show that in Colombia, the control of TB mortality in children under 15 years of age remains a challenge, and interdisciplinary, intersectoral, and intersectional actions focused on children and adolescents are required to improve outcomes in child survival. Similarly, tuberculosis in these age groups should be monitored as a sentinel event of the recent transmission of infection in the community, requiring high-level political commitment and the continued education of health personnel.

## 2. Materials and Methods

The study was conducted in three phases: (a) identification of the sociodemographic and clinical characteristics of the deceased subjects; (b) construction of indicators by department (mortality rate and YLL); and (c) stratification by epidemiological situations according to mortality rates and YLL.

A descriptive epidemiological study was conducted based on retrospective information from Colombia, corresponding to the period 2010–2018. The non-fetal mortality data were obtained from the Vital Statistics System, and validated by the Ministry of Health and Social Protection of Colombia, after consultation with that entity. For the identification of tuberculosis as a cause of death, the basic cause of death reported on the death certificate according to the IEX codes related to the health event was used. The inclusion criterion of all mortality cases reported in the country during 2010–2018 in people under fifteen years of age with a basic cause of death from any type of tuberculosis was used. The exclusion criteria were: fetal death records, duplicate records, and records in which the place of residence did not correspond to a department of Colombia.

Based on these criteria, 260 records were selected that fit the characteristics required for the analysis. A quality control process was carried out to detect typing errors and lost data, and two new databases were built, one for each unit of analysis: one with the people whose sociodemographic characteristics were taken into account and another with the departments in which the mortality rate and YYL indicators were constructed. No sampling was required because all the records in the database were taken into account.

### 2.1. Variables Studied

The sociodemographic variables of age, age group, sex, and origin were taken into account. The clinical variable was a basic cause of death, which for bivariate analysis was reclassified to tuberculosis type (pulmonary/extrapulmonary).

### 2.2. Statistical Analysis

Phase one was the identification of the sociodemographic and clinical characteristics of the deceased subjects. The variables were described by descriptive statistics (qualitative variables were expressed in frequencies and percentages, and quantitative variables were expressed in measures of central tendency and variability). The SPSS™ program (Bogotá, Colombia), version 23 was used.

Phase two was the construction of indicators for the country and by department (mortality rate, YYL, and YLL rate). The indicator of mortality rate, YLL, and the YYL rate were constructed for each department and for the study period using the number of cases per department as a numerator and the departmental population for the period according to the report of the National Department of Statistics (DANE) as a denominator. All rates were calculated using a constant of 100,000 inhabitants.

The data were obtained from the non-fetal mortality base of the Vital Statistics System of Colombia and validated by the Ministry of Health and Social Protection (MHSP). The consultation requested and carried out by the MHSP did not allow the identification of patients. For the identification of tuberculosis as a cause of death, the basic cause reported on the death certificate according to the ICDX codes related to tuberculosis was used. All forms of tuberculosis were included according to sex and age allowing control of garbage codes (ICD-X codes A150, A151, A152, A153, A154, A155, A156, A157, A158, A159, A160, A161, A162, A163, A164, A165, A167, A168, A169, A170, A171, A178, A179, A180, A181, A182, A183, A184, A185, A186, A187, A188, A190, A191, A192, A198, A199, J65X, K230, K673, K930, M011, M490, M900, N330, N740, N741, O980, P370). Based on the above criteria, a quality control process was carried out to detect typing errors and data loss.

In the Colombia context, the collection of data from the primary source begins when the attending physician generating the information according to the health outcome on the death certificate or the Individual Registry of Provision Services. Through the notification network, the Individual Registry of Provision Services is sent online from the Health Service Provider Institutions and the Benefit Plan Administration Companies. The Individual Registry of Provision Services enters the databases of the MHSP, where the information is available in the Social Protection System data warehouse. On the other hand, mortality records are generated from the institutions providing the health services and the National Institute of Legal Medicine and Forensic Sciences, with the system being in line with the municipal and departmental health directorates. The Single Register of Affiliates and the National Department of Statistics record mortality. The Independent Registry of Service Provision and the mortality registries periodically feed the affiliate data integration platform and the management web portal from which the national and territorial levels carry out the analyses.

For the YYL, the methodology of estimating the years lost due to premature death was utilized using the standard life expectancy according to the Global Health Estimates of the World Health Organization (WHO) [16] and the national population projections for the years 2005 to 2020 of the National Department of Statistics (DANE) by year, sex, and age group. The number of tuberculosis mortality cases was estimated for each territorial unit according to sex and age group (0–4 years, 5–9 years, and 10–14 years). We proceeded to calculate the YYLs using the following equation which comes from the simplified *YYLe*:$$YYL_{e}=\sum_{X=0}^{L}dx\;e_{x}^{*}$$
where (*L*) is the ideal standard life expectancy age, *dx* is the number of deaths in the population at age *x*, and $e_{x}^{*}$ = PF or life expectancy at each age (based on the standard life expectancy ELE).

The number of mortality cases according to cause, age, and sex was obtained, and the coefficients of the weighting factors by age of the standard life table were taken. The YYLs and their distribution by sex, age, and territorial unit were calculated. Using the SLSTAT 2021 program, 95% confidence intervals (95% CI) were established for the sum of the YYLs using the Bootstrap method. Then, 95% CI were calculated for each year of the study period by sex and year, by sex and age group, and at the subnational level for the study period. With Epidat 4.1, the crude and adjusted YYL rates (direct method) and 95% CI were calculated using the estimated YYLs for each of the nine years of study, the DANE population projection, and the constant 100,000.

Once the YYL rates were established, the median was calculated to define a comparison value from which the 75th percentile of these data was calculated, and two categories were defined: high mortality (entities with a value equal to or greater than the 75th percentile) and low mortality (entities with a value lower than the 75th percentile). The stratification according to high or low load (taken as YYL) was based on what was contemplated in the strategic plan “Colombia free of tuberculosis, 2010–2015, for the expansion and strengthening of the strategy ‘Stop tuberculosis’” [17] adopted in Colombia by the then Ministry of Social Protection.

Phase three was the stratification into epidemiological situations. According to the mortality rate and VPA rate, the premature mortality indicator was constructed, and the departmental entities were grouped as follows: those whose average was 90% or more of premature mortality and those with an average of less than 90%. In the field of tuberculosis control, the Millennium Development Goals set a target of reducing mortality by 90 per cent of those affected [18]. Based on the mortality rate (high or low) and the YYL mortality rate (greater or less than 90%), four situations were constructed in which the departments were stratified.

### 2.3. Multivariate Analysis

Crude and adjusted prevalence ratios and their 95% confidence intervals (95% CI) as well as the chi-square statistical test were used to establish whether the hypothesis that the rate of YYL (dependent variable) in people who died of tuberculosis had an association with the variables of sex, age group, and type of tuberculosis was true. A *p*-value less than or equal to 5% was taken into account.

### 2.4. Bias Control

Competitive risk selection bias was controlled by the theoretical assumption of attributing mortality to the basic cause; the bias of health care was mitigated with the inclusion of all reported cases covering the health care provider network of all territorial units along with the reports of the Colombian Institute of Legal Medicine. Information bias by underreporting could not be adjusted for due to the lack of measures of completeness of the vital statistics system [19]. The basic cause of death corresponded to the disease or injury that initiated the chain of pathological events that led directly to death, which for our study is tuberculosis. This cause was determined by reviewing the causal chain reported in the medical certificate of death and contrasting it with the basic cause codified by the National Department of Statistics.

### 2.5. Ethical Considerations

This study complied with all the requirements of Resolution 8430 of 1993 for health research in Colombia. The method of collection was documentary, no names or numbers of identity documents were used in order to safeguard confidentiality. In addition, the authors committed themselves to the policy of environmental protection and, in particular, to the rational use of resources.

## 3. Results

To facilitate their interpretation, the results are presented according to the phases developed.

### 3.1. Phase One—Identification of the Sociodemographic and Clinical Characteristics of the Deceased Subjects

In total, 50% of people under 15 years of age who died from TB (Figure A1) were 7 years old or younger (range 12 years); 65% were male, which was the sex that predominated in each age group. Of the age groups, those with the highest contributions were those aged 0–5 years (42.7%) and those aged 10–14 years (45.8%). Of the deaths, 32.3% came from Valle del Cauca and Antioquia. In terms of the disease, 66.5% of the cases were pulmonary tuberculosis (97.7% without bacteriological confirmation). Among extrapulmonary tuberculosis, the leading cause of death was tuberculous meningitis (12.7%).

### 3.2. Phase Two—Construction of Indicators for the Country and by Department (Mortality Rate, YYL, and YYL Rate)

In Colombia during the study period, the loss of 143,781 (95% CI 142,811–168,333) YYL was recorded in children under 15 years of age. The highest mortality rate in Colombia occurred in 2010 (0.6); the highest adjusted rates of YYL occurred in 2013 in the three age subgroups. The behavior of the YYL rate was fluctuating with a tendency to decrease.

Caquetá, Quindío, San Andrés, Guainía, Vaupés, and Vichada reported no cases during the study period. The departments in which tuberculosis mortality cases were concentrated were Valle de Cauca (17.7%) and Antioquia (14.6%); these two territorial entities contributed 37.8% of the YYL of the study period. The highest mortality rates occurred in Amazonas (10.2) and Chocó (5.8); the highest number of YYLs by age group was in children under 10–14 years (110,057). The highest rates of YYL by age group and territorial entity were: in children under 5 years of age in Amazonas (2659) and Chocó (2514); in the age group between 5 and 9 years in Arauca (1643) and Caldas (1642), and in the age group between 10 and 14 years in Amazonas (9671) and Guaviare (6681). Amazonas had the highest adjusted rate of YYL in the country during the study period (3979). In total, 36.4% of the territories had a high mortality.

### 3.3. Phase Three—Stratification in Epidemiological Situations

Regarding stratification by epidemiological situations, 30.3% of the territories corresponded to situation 1, 6.1% corresponded to situation 2, and 66.7% corresponded to situation 3. In total, 18.2% of the territories could not be located in any of the situations since they did not report cases (Figure A2 and Figure A3, Table A1).

### 3.4. Multivariate Analysis

Dependence was found between a mortality greater than 90% and the variables of sex and type of tuberculosis (Table A2).

## 4. Discussion

This study estimates for the first time the premature losses of years of life due to TB in children under 15 years of age in Colombia, showing that childhood tuberculosis remains a public health problem. According to age, a large percentage of cases occurred as pulmonary tuberculosis in children between 0 and 5 years and 10 and 14 years, a result consistent with Duarte et al. who determined the existence of a bimodal pattern of risk in these two groups in Portugal [20], and Llerena et al. where they observed that 62.4% of the cases belonged to these same age groups in a study in Colombia [21]. The median age was lower than that found in a regional study conducted in Caldas [7] in which it was 10 years, perhaps because most of the literature on tuberculosis divides the population into two age groups: “children” from 0 to 14 years and “adults” from ≥15 years. The highest number of mortality cases occurred in children aged 10–14 years, which is consistent with other studies [7,22,23], and although the reasons are not fully understood, increased susceptibility, sexual hormonal changes, and changes in social and immunological contact patterns have been established.

Pulmonary tuberculosis was the most frequent type of the disease, which coincides with results from Uganda [24] and Colombian regional studies [7,21] whose explanatory route may be due to the fact that the lungs continue to grow in volume and in the development of gas exchange capacity which can make children especially vulnerable [23,25]. It is important to note that, historically, age has been one of the co-principals in the evaluation of lung function, which is reached between 20 and 25 years. The most affected variables are FVC and FEV1, which present a more pronounced fall between 3 and 10 years apparently due to the gradual loss of lung elasticity [26]. In the study by Xing et al., it was reported that lung function parameters were worse in people who had tuberculosis, especially if this deterioration in lung function was not diagnosed or treated, which can result in increased mortality [27].

The high percentage of bacteriological non-confirmation is consistent with that reported by Yerramsetti et al. [28], which may be due to the difficulty of the collection of sputum samples and the very low bacterial load. The most frequent form of extrapulmonary TB was meningeal TB in contrast to what is described in the literature where the predominance of pleural and nodal TB is mentioned [29,30], a situation that may be due to the greater vulnerability of severe forms in children under 5 years of age or to the fact that TB is often overlooked as an underlying cause or comorbidity in children with pneumonia, malnutrition, or meningitis.

There are no previous studies that allow comparing the behavior of the rate of YYL and TB in Colombia. According to the Global Burden of Mortality Study from Tuberculosis in Children, this disease caused around 239,000 deaths in children under 15 years of age in 2015, more than 80% in children under 5 years of age, and more than 96% in children who did not receive antituberculosis treatment, with the region of the Americas being the one with the lowest rate. Some deaths from this event were probably not represented in the global estimates; so, the general estimates are low, which could explain the low rate of the region and the fluctuating behavior of the rate of YYL with a tendency to decrease with the country having a high burden of TB.

In addition, since the symptoms of TB are not specific, they may have been attributed to other diseases, including HIV infection [31]. The global burden of disease study mentions that the annual number of deaths from tuberculosis is decreasing globally; however, it is not fast enough to reach the 35% that the Sustainable Development Goal has as a milestone for 2020 [3]. The COVID-19 pandemic is a threat to reducing the global burden of TB disease; so, the global number of TB deaths could increase due to the disruption of health services, as well as the economic impact of the pandemic that has worsened social determinants such as GDP per capita and malnutrition [3].

The subnational differences are marked, striking is the non-reporting of cases in remote regions of high rurality and with differential populations such as Guainía, Vaupés, and Vichada, perhaps due to the geographical and administrative barriers impeding access to health services [32], low uptake, difficulties for access, high rurality, and migration which impose a programmatic challenge. Amazonas and Chocó showed high mortality rates and YYL in children under five years of age, which may be due to the adverse social conditions present that have a negative impact on the control of tuberculosis in children, as has been established in other Colombian regions with similar social conditions [33].

Depending on the Colombian region, the mortality of the disease can vary considerably, which may be due to the unusual way in which the disease appears in these particular groups, and is a situation also referenced in Brazil [34]. Studies related to the spatial distribution of TB cases in children show that they occur in regions with ethnic, racial, and immigrant minorities [35], and the particularities of the climate in which the cold decreases the performance of outdoor activities, limits access to fresh air, and increases the number of people inside the home, and dry weather could cause the late increasing trend of TB morbidity. Other factors may include educational conditions such as studying in boarding schools [36], vitamin D deficiency, economic conditions, the structure of the health and social protection system, characteristics of the population, or the conditions of each region, the analysis of which was not included in this study.

We found territories with notifications of mortality cases surrounded by others without the occurrence of cases, which may indicate underreporting and suggest centralization in some territorial entities and underreporting in their departments of origin, exhibiting a situation similar to that found in Brazil [37] where childhood tuberculosis was concentrated in the regions of metropolitan areas and urban centers.

Although information on the determinants of pediatric disease is scarce, TB–HIV co-infection, overcrowding, malnutrition, and oncological disease have been mentioned [34,38], among others. The work of the SDH results in the unequal and unfair distributions of the health of the population, among which are socioeconomic inequalities, mobility, urbanization, population growth, food insecurity, malnutrition, inadequate housing and environmental conditions, and cultural, financial, and geographical barriers to accessing health care [39] causing differential exposure and susceptibility to the disease and its severity.

The definition of epidemiological scenarios allows the territories to be grouped in such a way that they define and select interventions that can have an impact and bring greater benefits to the country in the control of TB. According to the present study, most of the departments that reported cases of mortality in the period are in situations 1 and 3. Situation 1 is characterized by weaknesses in the follow-up of patient treatment; situation 3 is characterized by some weaknesses in its information system. Weakness in treatment follow-up in patients affects treatment success, leading to a worse outcome [40].

Regarding dependence between the variables of sex and type of TB with mortality, it is important to bear in mind that in children the gender ratio is approximately 1:1. The findings described here are discordant regarding dependence on sex with the study by Llerena et al. [21] and consistent with what was found in this same research regarding age, perhaps because the risk of becoming sick is higher at very early ages but decreases between 6 and 10 years of age and then increases again. These findings are consistent with studies conducted in Pakistan and Brazil [34,40] in which there is dependence, although it is not clear whether they are due to biological factors or gender inequality in access to food, health care, and education.

The possible limitations of this research may be related to the use of secondary data. For the calculation of YYLs, it is not necessary to control the competitive risk due to the theoretical assumption that mortality is attributed to the basic cause of mortality. To control the bias of access to health care, all cases of mortality from the network of health care provider institutions in each territorial unit, from the different forms of participation in the health system and from legal medicine who report cases occurring outside the hospital network, were included. The misclassification caused by errors in diagnostic tests cannot be adjusted; so, this study assumed it. Because there is a specialized health program for tuberculosis, the Health Ministry was asked to cross-reference the information with the program.

Other comorbidities, such as immunodeficiency, diabetes, and hepatitis are expected to be extremely low in this population, and HIV testing is not routine. We also could not include social factors, such as household financial status or parents’ education level because they are not registered. Comorbidities and social factors may have had a direct or indirect impact on the child’s health or parents’ attention-seeking behavior, which may have contributed to the child experiencing failed treatment outcomes that led to death.

It can be said that this study brought together for the first time in the country a subnational approach to the estimation of YYL which can contribute to a better understanding of the behavior of the effects of TB in terms of early mortality as well as the approach to public health needs, related public policies, and their management. Additionally, it identifies priority sites for disease control.

Our results suggest that this country needs to evaluate data quality improvement processes in information systems and deploy strategies to identify children earlier in the disease process. These strategies may include active case finding, screening for household contacts of people diagnosed with TB, preventive treatment, community-based interventions to detect TB in the community or in areas where people may be at high risk, education campaigns to increase awareness about this disease, systematic monitoring of patients, and specialized pediatric teams that allow an accurate diagnosis to be made even in the absence of microbiological confirmation. In future studies, it will be necessary to address contextual, cultural, and regional situations that go beyond the analytical approach and the proposed design to deepen the explanation of the results.

## Data Availability

The data were obtained from the non-fetal mortality database of the Vital Statistics System (of Colombia, validated by the Ministry of Health and Social Protection of Colombia, upon consultation requested and carried out by the Ministry that does not allow the identification of the deceased.) The information can be consulted at https://orfeo.minsalud.gov.co/orfeo/consultaWebMinSalud/ (accessed on 15 January 2022), file No. 202142301807932 dated 23 September 2021, time: 03:07:03 and verification code 412cc.

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
