# Peer review of "Tuberculosis Mortality in Children under Fifteen Years of Age: Epidemiological Situation in Colombia, 2010–2018"

_tropicalmed, 2022, doi:10.3390/tropicalmed7070117_

Round 1
Reviewer 1 Report
Section methodology
Lines 118–124: Please, clarify how the authors worked with the garbage codes (GC) and why they did not use the associated cause of death.
Lines 158–162: As a suggestion to clarify how the authors elaborated the adjusted models. Did they consider variables with p-value less than or equal to 5% or did all variables enter the model? Remove the item "bivariate analysis" because the models were multivariate
Author Response
Dear Dr.
We appreciate the comments made after the review of our article. We attach the response to each of the comments. Please see the attachment.
Regards.

Reviewer 2 Report
Thank you for the opportunity of reviewing this paper. As known, our understanding of the TB epidemic in children is incomplete due to challenges in diagnosis and reporting, thus the analyses of death toll are relevant for better understanding the TB burden in this age-group.
However, the text needs some thorough revision before considering it for publication.
Please, find here other specific comments and suggestions.
Abstract
In general, the abstract is not clear and must be completely revised for readability.
Line 17: (range12) – This is not a range
Line 18: please, amend 14.3781. And are these YLL? Not clear
Introduction
Tuberculosis (TB) is the most important infectious disease in the world: how is this importance defined?
Methods
Authors may want to add some more context on Data sources (how data are collected and so on), diagnosis codes considered, possible garbage codes, and other important methodological aspects that allow readers to better understand the Colombian context.
Line 166: In Bias control, authors stated “Competitive risk selection bias was controlled by the theoretical assumption of at-165 tributing mortality to the basic cause”. It is not clear what a basic cause (or causes) is, and how it (they?) is determined? Did authors refer to the most frequent death causes in this age group in the country?
More on this analysis should be added too.
Discussion
Line 231 “whose explanatory route may be due to the fact that the lungs continue to grow in volume and in the development of gas exchange capacity, which can make children especially vulnerable”. This sentence needs citation, and authors should add something more on the mechanism by which the lung grow correlates with TB increased risk (the increase is in incidence, severity of disease????)
Author Response

(The authors gave the same response as above.)

Reviewer 3 Report
Estimated Authors of the paper "Tuberculosis mortality in children under fifteen years of age: epidemiological situation in Colombia, 2010-2018"
I've read your article with great interest. Despite its design, the present report may provide some significant information about the TB-related mortality in children (<15 y.o.) from a middle income country of South America, that is also characterized by substantial heterogeneity in socioeconomical features across the various territories and also within the very same regions.
From my point of view, the present study may be of some interest for the readers of TropicalMed and also for other professionals interested in Public Health and infectious diseases.
However, some substantial improvements are forcibly required before the eventual acceptance of this paper, mostly of formal nature.
First of all:
data reporting may be supported by a better referral of data in Figures and Tables. Table A1 and A2 contain data that are extensively reported across results section, and cannot be therefore considered as Annex ones.
Second:
statistical analysis may be improved by the implementation of I2 statistics, and calculation of heterogeneity across the various regions.
Third: are data from a time period before 2010 available? in case, I would suggest to implement the calculation of Excess Mortality Rate (EMR = (RDi − EDi,a)/EDi,a; where RDi,a = reported deaths in a given month i, at a specific administrative level a; EDi,a = average deaths in a given month i for the time period, at a specific administrative level a) may contribute to radically improve your calculations.
Author Response

(The authors gave the same response as above.)

Round 2
Reviewer 3 Report
Estimated Authors,
I've appreciated your efforts to improve the present paper, with some remarkable achievements.
In facts, only two minor remarks:
- please, move figures and tables across the text in order to improve the readability
- please double check the main text for some minor typos, that still affect the text (not extensively, but a minor editing is required)
Both improvements can be performed in editorial stage, therefore I'm advocating the acceptance of this study.
Author Response
Dear Doctors,
Thank you for your remarks. We attach the answer to each of them.
Regards
